# Exploiting Structural Modelling Tools to Explore Host-Translocated Effector Proteins

**DOI:** 10.3390/ijms222312962

**Published:** 2021-11-30

**Authors:** Sahel Amoozadeh, Jodie Johnston, Claudia-Nicole Meisrimler

**Affiliations:** 1School of Biological Science, University of Canterbury, Christchurch 8041, New Zealand; sahel.amoozadeh@pg.canterbury.ac.nz; 2School of Physical and Chemical Sciences, University of Canterbury, Christchurch 8041, New Zealand; jodie.johnston@canterbury.ac.nz

**Keywords:** effector proteins, fungi, oomycetes, protein modelling, RoseTTafold, AlphaFold2

## Abstract

Oomycete and fungal interactions with plants can be neutral, symbiotic or pathogenic with different impact on plant health and fitness. Both fungi and oomycetes can generate so-called effector proteins in order to successfully colonize the host plant. These proteins modify stress pathways, developmental processes and the innate immune system to the microbes’ benefit, with a very different outcome for the plant. Investigating the biological and functional roles of effectors during plant–microbe interactions are accessible through bioinformatics and experimental approaches. The next generation protein modeling software RoseTTafold and AlphaFold2 have made significant progress in defining the 3D-structure of proteins by utilizing novel machine-learning algorithms using amino acid sequences as their only input. As these two methods rely on super computers, Google Colabfold alternatives have received significant attention, making the approaches more accessible to users. Here, we focus on current structural biology, sequence motif and domain knowledge of effector proteins from filamentous microbes and discuss the broader use of novel modelling strategies, namely AlphaFold2 and RoseTTafold, in the field of effector biology. Finally, we compare the original programs and their Colab versions to assess current strengths, ease of access, limitations and future applications.

## 1. Introduction

Plants and their associated microbes have been interacting with each other for millions of years. Microbes can have either positive (mutualistic), neutral (communalistic), or deleterious (pathogenic) impact on plant fitness [1,2,3]. Plant–microbe interaction is a highly dynamic process not only affected by the interaction partners, but also the change of living conditions associated to abiotic factors affecting overall plant health and performance [3,4]. Symbiotic interactions can accelerate plant growth via nutrient acquisition like phosphorus and nitrogen or enhance plant resilience against various stresses [5,6,7]. Beneficial microbes are able to deploy tactics such as stimulating the plants immune system to generate anti-pathogenic products or activate defensive pathways [8]. The means of plant–microbe interaction are the same for symbiotic and pathogenic microbes, all evade or recruit methods to suppress the plants immune responses but with very different outcomes. Evidence has illustrated that endophytic beneficial microbes, similar to pathogens, combat with the plant’s defense layers [9,10].

Interestingly, the plant innate immune system does not distinguish between friend and foe, with both bringing about defensive processes collectively termed microbe-associated molecular patterns (MAMP) triggered immunity (MTI). This first layer of the immune response involves recognition of MAMPs (e.g., ergosterol, bacterial flagellin, Pep-13, xylanase) by so-called “Pattern Recognition Receptors” (PRRs) [11,12]. Invader induced damage to the plant can also induce the innate immune system via damage-associated molecular patterns (DAMPs). Hereby plant cell wall fragments can function as endogenous elicitors and serve as apoplastic signals to induce the immune system [13]. MAMP or DAMP activation of the PRRs eventually leads, via signaling cascades, to active defense responses, including production of reactive oxygen species (ROS), callose deposition, activation of the MAPK cascade and biosynthesis of jasmonic acid (JA) and salicylic acid (SA) [14]. MTI and pre-existing chemical and physical barriers comprise the plants basal resistance to most pathogens and microbes.

In order to establish synergy or to combat defense responses microbes may instigate further modulations. Adopted pathogens and mutualistic microbes evolved effector proteins that can either be secreted to the intracellular space or translocated into the host cell. In case of pathogens, suppression of MTI and the plant immune system is called effector triggered susceptibility (ETS) [15,16]. In contrast to pathogens, mutualistic and symbiotic microbes deploy effector proteins and secondary metabolites to establish symbiosis via root or shoot colonization [17]. To avoid ETS, host plants utilize the second layer of innate immune response called effector triggered immunity (ETI) [18]. The effector proteins and/or effector target complexes are recognized by the protein products of plant resistance genes (R-genes), which belong to the intracellular nucleotide-binding leucine-rich repeat (NB-LRR) protein family [19]. NB-LRR proteins are activated either through direct recognition of an effector or indirectly when the effector targets one of the host’s proteins. Their activation leads to ETI continued by an amplified disease resistance response [19]. Due to their key role during the infection process this review will focus on effector proteins originating from fungal and oomycete species.

## 2. Effector Proteins in Plant–Microbe Interaction

Phytopathogens and mutualistic microbes like filamentous fungi and oomycetes secrete effector proteins in order to colonize the host. Bioinformatic approaches allow us to predict effectomes from genome and RNA sequencing data sets, which is based on known signal peptides, effector motifs and domains (see below Section 2.2). Ultimately, these data sets are the starting point to explore effectors and their biological and biochemical functions. Depending on the host-range (specialist or broad range), fungi and oomycetes have both highly conserved and unique effector proteins. This allows them to target critical mechanisms involved in the plant’s immune system, stress adaptation pathways and regular cellular functions important to establish and survive on the host plant [20,21,22]. In the past decade, effectomes have been published for a wide range of pathogenic species, including dieback inducing species, rusts and smut fungi [23,24]. Recently, effectome sets are increasingly published for symbiotic and mutualistic microorganisms, allowing us to compare the molecular colonization strategies of pathogens and mutualistic microbes [25,26].

The most important function of effector proteins is the suppression of the plants signal transduction pathways associated with MTI, ETI or both [27,28]. To allow for their molecular function, effector proteins are either secreted into the apoplastic space or translocated into the host cell [29,30]. Hence, effector proteins are classified as apoplastic and cytosolic effector proteins (Figure 1). Apoplastic effectors can initiate the plant’s immune responses by early recognition in the plant–microbe interaction, but originally, they aim to induce successful colonization by blocking enzymatic reactions, mimicking plant proteins or disguising infection structures [31]. On the other hand, cytoplasmic effectors are delivered into plant cells to target intracellular processes, where they can be recognized by intracellular receptors [32]. In this review, we will focus on host translocated effector proteins of filamentous microbes, their known conserved domains, motifs and progression in the field of effector protein biochemistry and effector protein modelling.

### 2.1. All Lifestyles of Filamentous Microbes Use Effector Proteins to Establish Colonization

Oomycetes and fungi are filamentous eukaryotic organisms. In contrast to fungi that contain species of symbiotic and pathogenic lifestyle, oomycete species are mostly limited to a pathogenic lifestyle. Nevertheless, some oomycetes of the *Pythium* class are considered beneficial to plants and are in use as bio-control organisms. Examples are *Pythium olingandrum* and *Pythium periplocum*, which are known to be mycoparasites that antagonizes fungal plant pathogens [33,34].

Until recently, effector proteins have been studied mainly in context with pathogenic fungi and oomycete species. Nevertheless, recent advances have shown that symbiotic organisms such as endophytes and mutualistic microorganisms also secrete effector proteins [35,36] (Table 1). According to Rovenich et al., 2014 effector proteins contribute to niche colonization and most likely to microbial competition [29]. In mutualistic connections, identical to pathogenic invasions, the microorganism is identified by the plant’s recognition system and needs solutions to evade the plant’s immune strategies to maintain a mutual beneficial connection [37]. Apoplastic secreted effectors, such as secreted proteins (SP’s), β-glucan, [38] or RiSLM that binds to chitin [39], are known to play a role in early establishment of mycorrhiza–plant interaction. Recently, effectors translocated into the host’s cytosol originating from symbiotic fungi become more and more the focal point of ongoing research (Table 1) and we start to understand that oomycetes and fungi of all lifestyles use effector proteins to establish an interaction with the host plant [40,41]. This includes translocated effector proteins containing RxLR motifs and crinklers (CRN’s), which will be reviewed in more detail in the next chapter. Effectors are likely to be used by plant growth promoting fungi to limit the activation of the plant’s immune system by decreasing the amount of specific MAMPs recognized by the plant’s PRRs. However, many questions remain unresolved about the molecular mechanisms governing mycorrhiza–plant interaction—with one being how they can establish interaction with such a broad host spectrum. Future research in this field will need to establish collaborative approaches, combining ecology (bigger picture), molecular interaction studies of microbe and host on the cellular level (organismal and cell level) and protein biochemistry approaches (molecular level) to resolve these important questions.

### 2.2. Effector Proteins of Filamentous Microbes

Most of our knowledge on effector protein function, motifs, domains and structures derives from pathogenic species rather than beneficial and symbiotic species. Compared to oomycetes, identification of motifs and domains involved in delivering cytoplasmic effectors has been particularly challenging for fungi due to less clear sequence conservation. Nevertheless, fungi and oomycetes have been shown to translocate RxLR/RxLR-like effectors and CRNs into the host cell [41]. Oomycetes contain a particularly high number of RxLR effector proteins, which are likely to be secreted via the haustoria during plant–oomycete interaction [49]. RxLR effector proteins are composed of an N-terminal signal peptide responsible for effector secretion, followed by a highly conserved RxLR (Arg-Xaa-Leu-Arg) motif. This motif has been proposed to be in charge of the translocation of the effector protein into the host cell [50,51]. More recently, it has been hypothesized that the RxLR motif is cleaved before translocation into the plant cell and only a mature effector protein containing the C-terminal effector domain is delivered into the host cell [52]. The RxLR motif is often followed by a downstream (D)EER motif (Glu-Glu-Arg) located within 40 AA after the signal peptide, which is also linked to the effector translocation [50,53]. The effector proteins of *Phytophtora* species such as *P. infestans* (Avr3a and PexRD2), *P. capsici* (Avr3a11) and downy mildews such as *Hyaloperonospora arabidopsis* (Hpa; Atr1) also contain a WY or WL motif, which forms an alpha-helix [54]. The motif, identified by analyzing the crystal structure of PexRD2, is comprised of two hydrophobic residues buried inside the protein core that contribute to interactions with host target proteins. WY-containing effectors and their structures have been recently reviewed in detail by Mukhi, et al. 2020 [54]. Other RxLR effectors have been shown to interact with their targets in the cellular endomembrane system, including *P. infestans*’s effector protein Pi03912 and *Bremia Lactucae*’s effector proteins BLR05 and BLR09 that interact with NAC transcription factors located in endoplasmic reticulum [55,56].

Translocated CRN effector proteins are distributed in nearly all pathogenic oomycetes and have been shown to be translocated by fungi of pathogenic and beneficial lifestyle. CRN’s share two conserved motifs in their N-terminal region, the LxLFLAK (Leu-Xaa-Leu-Phe-Leu-Ala-Lys) motif and the HVLVVVP (His-Val-Leu-Val-Val-Val-Pro) motif. The LxLFLAK motif is, comparable to the RxLR, associated with the translocation of the effector in to the host cell [57,58]. CRNs, initially identified through their ability to cause crinkling and necrosis upon expression in plant tissue are not typified by this characteristic. In fact, expression of CRNs leads to cell death only in a select few cases. So far, CRNs are less well studied than RxLRs [57,58].

Fungal species have further effector proteins with various effector motifs including but not restricted to, lysin (LysM), DELD, RSIDEDLD, RGD and the EAR (ethylene-responsive element binding factor-associated amphiphilic repression) motif.

Furthermore, most MAX effectors (Magnaporthe AVRs and ToxB- like effectors) so far have been identified to be translocated, contributing to the virulence of pathogenic fungi. These effectors contain a β-sandwich fold, showing similarities to the apoplast secreted *Pyrenapohora tritici-repentis* ToxB. This group of effectors have at least one disulfide bond with variable AA on their protein surface, which mediates their target interaction [59,60]. RALPHs (Rnase-like proteins expressed in haustoria) are another group of fungal translocated effectors discovered in pathogenic fungi, including the *Blumeria graminis* effector BEC1054. RALPHs block the function of the host’s ribosome, inactivating proteins and suppress the host cell death [61]. The flax rust effector AvrP is considered an HESP (haustorial expressed secreted protein) that does not contain an RxLR and the translocation mechanisms in the host cell is not clear to date. Nevertheless, it is one of the few effector proteins with a known structure. It contains Zn-finger like motifs and three Zn- binding sites. The Zn-finger motifs are necessary for maintaining the integrity of the effector protein and cell death activity [62]. Other structurally resolved fungal and oomycete effector proteins are presented in Table 2.

Interestingly, even though filamentous effector proteins have been studied and defined extensively with genetic and molecular biology approaches, available protein structures are very limited (Table 2). Structural information is very valuable for elucidating the molecular mechanisms behind biological and biochemical functions. It is complimentary to genetic and molecular biology methods, giving a molecular explanation for observations seen in these studies and seeding hypothesis for further of these studies. In addition, the fundamental molecular level insights ultimately help link genome and sequence information to function and aiding improvements in effectome prediction. Considering the importance of effector molecules during infection processes of plants, but also of humans and animals, it is surprising that effector proteins have not been studied more intensively. This in part may be due to experimental challenges with structure elucidation, including the membrane-associated nature of many effector proteins and the potentially dynamic nature of their different molecular interactions along the infection/colonization cycle. Nonetheless, structures and their detailed molecular function, are a significant knowledge gap and that is true for oomycete as much as for fungal effectors.

## 3. Structures and Computational Modelling of Effector Proteins

To decipher the function and evolutionary pattern of different species, investigating the 3D structure of effector proteins can play a promising role. Across species effector proteins show only low sequence similarity, which limits the power of sequence-based analysis for predicting functional and evolutionary patterns. It is predicted that structural conservation of effector proteins may be able to resolve these evolutionary and functional relationships (shared, but with low conservation) that sequence analysis alone cannot define [64]. *Magnaporthe oryzae* effector proteins AVR1-CO39 and AVR-Pia advocate this statement. The NMR spectroscopy of these effector proteins has shown that even though they lack sequence similarity, they both contain six β-sandwiches in their structures with Cys disulfide bridges located in the same positions. AvrPiz-t and ToxB are two other effector proteins of *M. oryzae,* which are sequentially unrelated but possess structural similarities, forming the MAX effector family [59]. In addition, the power of large-scale structural determination should not be underestimated. The more structural information exists for different and distantly related effectors, the more sequences can be linked to molecular level function. This in turn aids better prediction of effectomes from sequence alone and seeds more experiments on function that feedback into the process. To support this point, the structural resolution and identification of the WY motif in effectors from filamentous microbes can be considered one of the key discoveries in the last decade, as it helped to classify a completely new class of effector proteins. Importantly, this motif was only discovered, because of the structural analysis of certain effectors Avr3a and PexRD2 [65], which underlines the importance of structural resolution and modelling in this field.

Analysis and prediction of 3D-structures of biological macromolecules such as proteins, DNA and RNA are studied within the field of computational biology. In recent years, computational modelling has become one of the most rapidly evolving research areas in structural biology [66,67]. It is a highly utilized area by those without expertise in structural biology, with clear reasons in public accessibility of many tools (online) and their utility to inform a wide range of experiments across biology. Experimentally, protein structures are mainly investigated through methods like X-ray crystallography, nuclear magnetic resonance (NMR), bio-SAXS and cryogenic electron microscopy (Cryo-EM). While powerful, with different strengths, each approach has limitations and involve considerable investments of financial resources and time. Even then, there is no promise a structure can be resolved, particularly for certain types of proteins, including those with intrinsic disorder or those residing or interacting with membranes. Nonetheless over the past nearly 50 years a wealth of structural information has been accrued, with near exponential growth, for the protein world which has been publicly deposited freely in the Protein Data Bank (PDB) [68]. Computational methods, like structural bioinformatics have developed as a result of, and benefitted greatly from, this public PDB resource and have in turn aided value from its contents. Developing structural analysis tools gain us comprehensive information about folds and local motifs in proteins, evolution and function/structure relationships and molecular folding, in order to understand the main functions of proteins and their role in biological processes [69]. Over the last decade scientists have evolved various systems and algorithms to overcome the 3D protein structure prediction problem [70]. Improvement in protein energy functions [71], protein conformational sampling and sequence optimization, as well as rapid growth in biological databases has made great advances in protein structure prediction [72]. We believe that the recent developments in the field of computational protein modeling will become of increasing value for the elucidation of plant–microbe interactions and effector biology.

## 4. Protein Modelling Approaches for Effector Proteins

Analyzing the structure of effector proteins is critical for our understanding of the molecular mechanism behind the pathogenic and symbiotic interaction processes. Meanwhile, characterizing effector genes and proteins in a genome-wide scale provides a great insight into their functional roles, classifying them based on their conserved sequence motifs and deciphering their evolutionary patterns [73]. Moreover, predicting and identifying effectors and their host target proteins, open up opportunities to realize related pathways involved, and ways of manipulating them to establish plant protection [74]. There are thousands of effector proteins known and many not known yet. Table 2 lists the known experimentally determined effector proteins, currently represented in the PDB; to our knowledge, 32 structures of fungal effector proteins and 12 structures of oomycete effector proteins are available in the RCSB PDB database at present [75]. Among the experimentally determined structures revealed in the PDB database, the effector proteins of bacterial origin were most dominant [54] with structures mostly resolved by X-ray crystallography (Table 2). As can be seen by the family distribution in the table, some effector types are represented (e.g., RXLR/WY proteins have numerous representatives) while other effector families are not represented at all. As discussed in the section above there are reasons for this lack of representation, including technical limitations. The last decade has witnessed extraordinary advances in computational methods, which have had great impact on the field of structural biology. This includes visualization of structures, data analysis and sequence to structure prediction. The prediction of effector structures via computational modelling mainly uses sequence-based approaches with machine learning and deep learning computational methods and has been recently reviewed in Suh et al. 2021 [76].

Depending on the amount of prior knowledge on related proteins with similar sequences, two structural modelling approaches have been established: (1) If related sequences have been structurally characterized with representatives in the PDB then homology or comparative modeling is the first method of choice, (2) in case of no known structures, ab initio or de novo modeling is the most common approach. Homology modelling has been implemented in bioinformatics prediction tools such as SWISS-Model [77], Modeller [78] and Phyre [79] (Protein Homology/analogy Recognition Engine), and relies on the assumption that similar sequences with common ancestors would probably possess similar structures [75,76]. It essentially comprises seven steps, starting with a template search and initial alignment-to-alignment refinement, model building comprising of backbone production, loop modeling and side-chain modeling, model optimization and validation [80]. Recent advances have been introduced to these steps of homology modeling, improving the overall output [81]. Homology modelling has proven very useful, particularly for proteins with high sequence identity, nevertheless, the method has a number of limitations. This includes the limited confidence with which it can predict the structural effect of point mutations and multi domain proteins and its accuracy waning with lower sequence identity, with sequence identity <30% being the lower limit for confidence in the model. In the situation where no sequence similarities are available in the data base or when the identity percentage is <30%, the protein structure prediction is directly constructed from scratch using the ab initio or de novo modeling methods [74]. The current technology progresses combined with advanced contact mapping (the map indicating the distance between the amino acid residues), co-evolutionary analysis empowered by state-of-the-art neural networks allows improved prediction of larger-sized protein folds [82]. Since 1994, novel structural prediction techniques are discussed and evaluated every two years at CASP (Critical assessment of protein structure prediction), a global protein structure prediction competition. Here, competitors apply their novel prediction tool(s) to solve yet-unpublished protein structures. However, they are exclusively provided with the amino acid sequences of the proteins. Results are compared to the corresponding experimental structures and similarities evaluated based on accuracy of the predicted models. Matrices commonly used for evaluation are the Global Distance Test Total Score (GDT_TS) and the TM-score (Template Modeling score) [83].

### 4.1. New Developments: AlphaFold2 and RoseTTafold

The two last editions of CASP, (CASP13 and CASP14) have yielded some new modelling procedures, including one approach, AlphaFold2 (developed by DeepMind technologies, London, Great Britain) that has been touted as breakthrough advance in the field [84]. AlphaFold2, revealed in CASP14, is a highly superior version of AlphaFold1 with the novel transformer initiative and repeated form of analysis. AlphaFold2 was introduced by the Google DeepMind team. They won the competition achieving an average GDT_TS of 85.1 [85], which was relatively higher than the maximum GDT score of 65.7 achieved in the CASP13 round [86]. The end-to-end AlphaFold2 prediction program used in CASP14 is generally comprised of two stages. Simply speaking, in the first stage the given amino acid sequence is used for constructing the multiple sequence alignment (MSA). In this step the query amino acid sequence is paired with multiple homolog sequences derived from different species and also with the individual residues, constructing *N*_seq_ × *N*_res_ and *N*_res_ × *N*_res_ (*N*_seq_: number of sequences and *N*_res_: number of residues) matrixes, respectively. Multiple template matrixes from structure database search are additionally used. The reason is the general conservation of protein structures despite mutational or evolutional sequence differences. The result of the two matrixes is the MSA representation and the pair interaction information. One of the breakthrough points of AlphaFold2 was the introduction of the evoformer or transformer, which is responsible for refining the mentioned representations by iteratively exchanging their information to come to a more precise conclusion. For instance, in the published model the evoformer had repeatedly refined the information with 48 cycles. In the second step, the neural network takes the MSA representation and pair representation information to construct a static structure module in just one-step unlike other novel models, which utilize many optimization procedures. The very exciting novelty of AlphaFold2 belongs to its refinement cycles, which iteratively use the output of the first stage (MSA representation and pair representation) and the output of the second stage (the predicted structure) in order to process them repeatedly between the evoformer and the final step in model prediction [87].

In AlphaFold2 the predicted model is evaluated by overlaying the original structures existing in PDB with an IDDT (local Distance Difference Test) score, which computes the overall score by considering all heavy atoms or IDDT-Cα which measures the backbone accuracy based on the Cα atom. RMSD_95_ (Cα Root-Mean-Square-Deviation at 95% coverage) is also reported as an accuracy metric in AlphaFold2 [87]. Root-mean-square deviation of atomic positions measure the average distance between the atoms, particularly the backbone atoms of the two superimposed proteins, and it is calculated in angstroms. The lower the distance in angstroms, the more similarity exists between the proteins [88]. AlphaFold2 takes advantage of graphics processing unit (GPU) depending on the number of the amino acids of the protein query. For instance, a V100GPU is able to predict a 256 residue query in 4.8 min, 384 residues in 9.2 min and 2500 residues in 18 h. When the ensembling option is off, all predictions are 8× faster; 0.6 min, 1.1 min and 2.1 h respectively. Nevertheless, the ensembling procedure has shown to have minor effects on the accuracy of the predicted models [87].

The other novel program utilized in the field of protein structure prediction and modeling, RoseTTafold, was also released in 2021 to the scientific community [89]. RoseTTafold is a software tool utilizing deep learning methods to predict protein structures computationally. This modeling software developed by RosettaCommons (developed by Baker Laboratory, University of Washington, United States of America) uses a three-track neural network, providing a better performance than trRosetta and Robetta, the older generation prediction tools. Despite the two-track AlphaFold2, RoseTTafold takes advantage of a three-track attention network comprising of (1) information from the 1D amino acid sequence, (2) the 2D distance map and (3) the 3D backbone coordinates. In all three steps information goes back and forth to generate an accurate structure. The 1D and 2D track are in the 2-track block and the 3D track forms the 3-track block. Furthermore, to increase modelling performance RoseTTafold relies on the implementation of a transformer function. For a protein query containing less than 400 residues RoseTTafold requires approximately 1.5 h for sequence and template search, and ~10 min on an 8G RTX2080 GPU for the end-to-end procedure to produce the backbone of the predicted model [90].

AlphaFold2 and RoseTTafold, are clearly two novel breakthrough approaches in the modelling space with a lot of potential, in particular for the prediction of effector structures. Potential integration in high-through-put and effectome prediction pipelines would allow us to gain more insight on 3D conservation of effectors and their potential targets in plants. Nevertheless, the full version of both AlphaFold2 and RoseTTafold need accessibility of intensive core computing facilities.

### 4.2. User-Friendly Colabfold Alternatives

The full versions of AlphaFold2 and RoseTTafold require super computers in relation to memory usage. Google Colabfold has made the access to AlphaFold2 and RoseTTafold much easier by providing free computer resources, namely powerful GPUs for machine learning applications. There are numbers of Notebooks provided by Google Colabfold, responsible for modelling protein structures, each having specific parameters involved but of course, not possessing all the algorithms used in the full version. AlphaFold2_mmseqs2, AlphaFold2_advanced, AlphaFold2_batch, AlphaFold2 (from Deepmind) and RoseTTaFold are the current available notebooks in Google Colabfold. Updates are published regularly, and users need to check the most up-to-date notebooks available. AlphaFold2_mmseqs2 is for basic users and predicts structures based on MSAs produced by MMseq2 or the MSA file uploaded by the user and when predicting the structures, it gives the option to choose from already existing experimental based templates, relaxing the structures using amber and generating up to five models. The AlphaFold2_advanced notebook on the other hand provides users with more advanced options such as constructing MSA using HMMer, number of random seeds (num_samples), number of times the structures go back to Evoformer for refinement (max-recycles) which has the option to choose from 1 to 48 cycles (3 recommended), number of ensembles (num_ensemble) and enabling the stochastic part of the model (is_training). Both notebooks generate a LDDT (Local Distance Different Test) plot of their predicted proteins showing the accuracy of different secondary formations with different colors. In both RoseTTafold and AlphaFold2 notebooks using templates has been shown to have a surprisingly minor effect on the quality of the predictions [91], hence, they allow users to produce high-quality protein models even without access to high-performance computing facilities.

### 4.3. AlphaFold2 and RoseTTafold for Structural Prediction of Effector Proteins

In this review, we have compared the full version of AlphaFold2, RoseTTafold and the presently available Google Colab version for the modelling of different groups of effector proteins, originating from filamentous microbes in order to assess their relative utility to the effector protein field (Figure 2). For practical reasons, we have chosen effectors with existing 3D-structures in the PDB database and compared modelling results with the experimentally resolved protein structures. This set of known structures shows diversity, encompassing different secondary structures including, α-helices, β-sheets and turns from both oomycete and fungal effectors. From these six chosen effector proteins, three are oomycete RxLR effector proteins namely PexRD2 [3ZRG], Avr1d [7C96] and Avr3a [2NAR] and three are non-RxLR fungi effector proteins including Avr2 [5OD4], ApikL2A [7NLJ] and AvrP [5VJJ]. The *P. infestans* Avr3a effector protein structure consists of a single three-helix WY domain continued by an extended N-terminal helix (the K motif) making a four-helix bundle overall. It has been observed that the fourth α-helix of this protein has an additional bend in comparison to other WY domains. The N-terminus of Avr3a forms a highly flexible domain ready for protease cleavage. In contrast the domain consisting of Glu-70 to Tyr-147 makes the rigid core of the effector protein [52]. PexRD2, another RxLR effector protein of *P. infestans*, also forms a single three-helix WY domain. It has been demonstrated that the third loop of the three α-helices in this protein is longer when compared to other existing RxLR effector proteins of the WY domain containing class. In contrast to Avr3a, which has a monomeric formation, PexRD2 self-associates and forms a dimeric functional mode of the effector protein in planta [92]. Avr1d (Avh6) is a *P. sojae* effector protein comprising of a signal peptide continued by a conserved N-terminal RxLR motif and an effector domain. Recent studies have shown its interaction with the U-box-type 3 E3 ligase of GmPUB13, functioning as a susceptibility factor for plants. The structure of this effector has been demonstrated to contain a single WY motif with a three α-helix bundle, in which the Tyr-118 and Trp-96 are able to form a hydrophobic core enabling the interaction with GmPUB13. It has been observed that Phe-90 is the prominent amino acid in this interaction, inactivating the ubiquitin ligase activity of GmPUB13 and contributing to the infection process [93]. In contrast, Avr2 is a non-RxLR fungal effector protein produced by *F. oxysporum*, comprising of a β-sandwich fold which is generated by two antiparallel β-sheets. The structure has been released in PDB and shows that disulfide bonds between Cys amino acids exist between different the β-sheets, stabilizing the β-sheets and associating the N-terminus of the β-sheets to the core region of the β-sandwich [94]. APikL2A is an allelic variant of APikL2 effector protein of *M. orzyae*. It has been shown to have binding capabilities to the host’s heavy-metal associated (HMA) domain family (similar to AVR-Pik)—vital for the *M. orzyae* infection process. APikL2A has been shown to have similarities to the MAX-effector protein class [95]. The last of the six effector proteins, AvrP, is translocated by the flax rust fungus into the host plant. It consists of four β-strands and one short α-helix at the C-teminus. Three Zn-ions have been shown to interact within these effector proteins. The virulence action of AvrP is associated to these Zn-binding motifs, which eventually induce cell death in the host plant [62].

By doing this systematic comparison, with a diverse range of known effector proteins, we are able to assess not only both programs that have been touted as major recent advances, but also determine if the easier and faster to run Colab alternatives produce models as accurate as their full versions. Moreover, it allows to observe which secondary structures may be predict better—or weaknesses of prediction set-ups. One way to assess the reliability of the two novel programs is to compare the accuracy of modelled secondary structures to the experimentally determined structures using overlays (Figure 2). Furthermore, RMSD and GDT-TS were used as similarity comparison metrics for evaluation of the models (Figure 2).

Overall, both full versions of the programs performed well, particularly across defined secondary structure regions, though each had differences in performance with different types of structure. Interestingly, in most cases the Colab version of AlphaFold2 has shown structures with similar RMSD values as the original full AlphaFold2 version. On the other hand, the RoseTTafold Google Colab version was not as successful as AlphaFold2′s Colab version. It had significantly lower RMSD scores than its full version, confirming the fact that the RoseTTafold Google Colab version might need more updates and newer versions. While the full AlphaFold2 version performed for some proteins slightly better than its Colab version (PexRD2, Avr3A)—overall performance was rather comparable between the two. Furthermore, modeling of proteins with β-sheets seemingly challenged both AlphaFold2 versions more than the full RoseTTafold version, at least for the modelled effector AvrP (Figure 2). It will be interesting to see, if future effector modelling approaches might further confirm this aspect of these new modelling approaches.

Similar to former programs, flexible long loops are still the most challenging structures to predict in all versions and might need to rely on experimental approaches for re-evaluation of the model. According to our comparison, we can say that the Google Colab versions have over all more positive than negative points. Besides their apparent benefits, including being user friendly and time- and cost-effective, their accuracy of model predictions in comparison to the original programs is not negligible. However, utilizing the full versions of the programs are able to give us more details in every aspect of the modeling parameters, the Google Colab versions can aid scientists to model their desired proteins—especially if they do not possess access to super computers. Nevertheless, best practice is to use multiple modelling approaches as shown here, in particular for proteins with completely unknown structures. This allows comparison of the resulting models and helps with the decision on whether the model is acceptable or not. In our experience, regions that are predicted similarly within different modelling tools have a significant chance to be correct. In contrast, regions that are predicted more divergently between different modelling tools, have a higher chance to be predicted “incorrect” or less precise. Furthermore, it is possible that less accurately predicted regions reflect high dynamics and potential functional importance, for example, sites involved in substrate-binding, protein–protein interaction or post-translationally induced conformational changes. In this context, the newly released AlphaFold-Multimer has been developed to predict in particular multi-chain protein complexes. Along prediction of multimeric interfaces, it allows high accuracy prediction of intra-chain interactions. This new tool will find application in protein-protein interaction research and facilitate future insights on effector-target interactions [103].

## 5. Conclusions

Plants are faced with a diverse range of microbiota—from the neutrals, beneficials to the pathogenic. Where pathogens cause devastating diseases, beneficial microorganisms enhance the plants adaptation to abiotic conditions and defense against pathogenic organisms, accelerating over all the plants fitness. These microbes are regularly recognized by the plant’s immune system by their MAMPs. Recent research shows that plants do not distinguish between friends or foes, generating highly similar defensive reactions. We have shown in this review that both pathogenic and beneficial filamentous microbes use so called “effector proteins” in order to gain access and establish an interaction with the host. In contrast to pathogenic effector proteins, not many studies have worked on the functional roles of effector proteins of beneficial filamentous microbes. For both pathogens and symbionts, availability of effector protein structures is limited, leading to a significant knowledge gap on conservation, function and interaction with target proteins. Ultimately, future research on structures and structure models of effectors and their corresponding targets will allow discovering new (conserved) motifs, domains, their evolution and role in plant–microbe interaction. Experimental procedures such as NMR and X-ray crystallography have made great advances in structural biology and effector biology, nevertheless they are still a limiting factor. The use of computational prediction methods can overcome some of these limitations and pertinently are more accessible to those in the field than time-consuming, expensive, expertise intensive experimental approaches. Despite this computational prediction methods have their own limitations, though recently, utilization of novel artificial intelligence has been able to counteract the lagging of protein prediction procedures. The most recent advances, AlphaFold2 and RoseTTafold, have great potential to support the field of effector biology with accurate modeling of their 3D structures. Their user-friendly Colab versions (now also available via the new version of ChimeraX [104]) will allow a broad user base to apply 3D-modelling for effectors and integrate them into existing effectome identification pipelines, which will allow for more precise identification of effectomes. Of particular interest will be the approach for the identification of unknown conserved 3D structures involved in effector-target interaction and the modelling of the interactions by multimer modelling [103].

In cases, where no experimental comparison is available, it will be important to have ways to validate the resulting predicted structures. Our observations have suggested that each program, and Colab version has different strengths and weaknesses. For proteins with unknown functions it will be very important to utilize, and compare the results of, more than one method for structure prediction. It is also important to be aware of the limitations of prediction methods and the fact that the protein structure is dynamic and one predicted structure may not represent all forms possible for an effector protein. Limitations aside the recent advances in ab initio modelling software have great potential to lead to effector structures that give molecular level insights and inform and inspire future experiments probing plant–microbiome interaction.

## Figures and Tables

**Figure 1 ijms-22-12962-f001:**
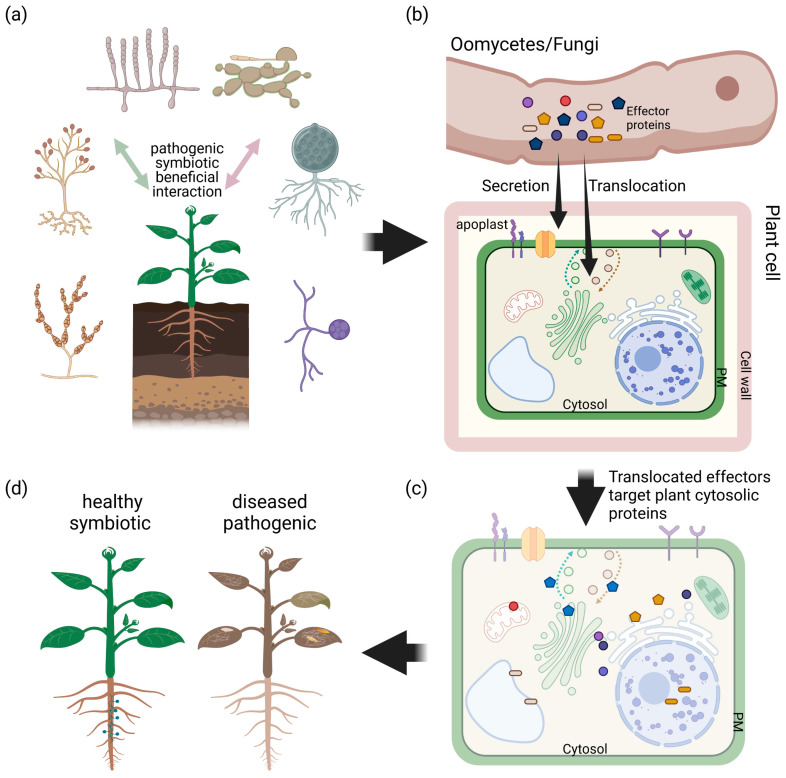
Symbiotic and pathogenic filamentous microbes us effector proteins with very different outcome for the host. This figure shows a simplified model for interaction of plants with filamentous microbes, which can be of beneficial, symbiotic or pathogenic origin (**a**). A key element of plant–microbe interactions is the secretion of effectors into the apoplastic space (apoplastic effectors) or translocation into the host cell (cytosolic effectors) by the microbe (**b**). It has been shown that beneficial as well as pathogenic oomycetes and fungi use conserved and unique species-specific effector proteins to modulate the host’s immune system targeting proteins in organelles, cytosol and intermembrane system (**c**), with a very different outcome for the plant (**d**).

**Figure 2 ijms-22-12962-f002:**
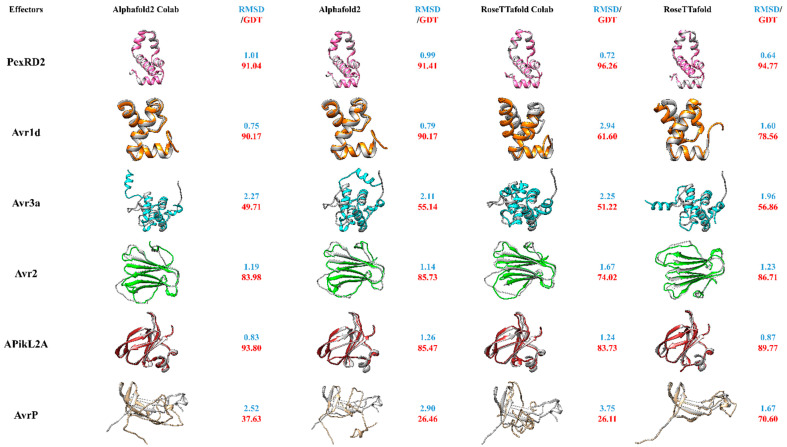
The predicted structures of PexRD2, Avr1d, Avr3a, Avr2, ApiKL2a and AvrP using AlphaFold2 Google colab version, AlphaFold2 full version, RoseTTadfold Google Colab version and RoseTTafold full version and their RMSD and GDT-TS values calculated when superimposing models with their PDB structures. The following methods were used to model the structures: AlphaFold2_advanced Google Colab notebook [96] with the settings of: MSA_method: MMseq2 (fast method), max_msa: 512:1024, num_models: 5, active use_ptm, num_ensemble: 1, max_recycles: 3, num_relax: Top1, NeSI (New Zealand eScience Infrastructure) AlphaFold2 full version [97], RoseTTafold Google Colab notebook [98] with the setting of: MSA_method: MMseq2, and RoseTTafold Robetta server [90] and the RMSD value (blue) of the superimposition of the proteins PDB structures and their corresponding models calculated by PDBefold [99,100]. GDT-TS (red) was calculated using the online LGA (Local-Global-Alignment) program [101]. UCSF Chimera version 1.15 [102] was used to visualize the models.

**Table 1 ijms-22-12962-t001:** List of effector proteins identified for beneficial fungi, their host species and biological function.

Effector Protein	Fungal Species	Host Species	Characterized Biological Function	References
SP7	*Glomus intraradices*	*Medicago truncatula*	Interacts with JA/ethylene inducible ERF19 transcription factor and down regulates PTI	[39]
Lysm effector Tal6	*Trichoderma atroviride*	*Arabidopsis thaliana*	Binds to chitin of plant’s cell wall and protects the fungi hyphae from plant’s chitinase favoring *Trichoderma* interaction and increasing mycoparasitic effect	[42]
Lysm effector RiSLM	*Rhizophagus irregularis*	*Medicago truncatula*	Binds to chitin and chitooligosaccharides of plant’s cell wall and interferes with chitin-triggered immune response protecting hyphae from plant’s chitinase and enabling symbiotic reactions	[43]
MiSSP7	*Laccaria bicolor*	*Populus trichocarpa*	Suppresses JA-mediated immune response by preventing JA-dependent degradation of PtJAZ6, a negative regulator of JA-induced genes	[44]
RiCRN1	*Rhizophagus irregularis*	*Medicago truncatula Nicotiana benthamiana*	Establishes a functional AM symbiosis and Arbuscules phosphate transporter gene-MtP4-expression	[45]
Strigolactone induced secreted protein 1 (SIS1)	*Rhizophagus irregularis*	*Medicago truncatula*	Essential for AM symbiosis, gene silencing causes suppression of colonization and production of stunted arbuscules	[25]
RP8598 and RP23081	*Rhizophagus proliferus*	*Medicago truncatula Nicotiana benthamiana Allium schoenoprasum*	Interacts with JA/ethylene inducible ERF19 transcription factor and down regulates PTI	[22]
Nuclear localizing effector (RiNLE1)	*Rhizophagus irregularis*	*Medicago truncatula*	Interferes with mono-ubiquitination of 2B histone and decreases the expression of defense-related genes while enhancing AM colonization process	[46]
Hydrophobin-like OmSSP1	*Ericoid mycorrhiza*	*Vaccinium myrtillu*	Mutants are unable to colonize *V. myrtillu* roots and OmSSP1 may strengthen the attachment of the fungi to the root protecting the hyphae from plant’s immune system	[26]
PIIN_08944	*Piriformospora indica*	*Arabidopsis thaliana*	Mutants show delayed colonization and PIIN_08944 expression reveals impairment of SA-defense pathway and reduced expression of flg-22	[47]
Did1 (PIIN_05872)	*Piriformospora indica*	*-*	Interferes with iron-mediated defense response which plays an important role in ROS generation	[48]

**Table 2 ijms-22-12962-t002:** Summary of structurally resolved effector proteins available in PDB-deposited structures [63].

Effector Protein	Organism	Date of Release	Method	PDB Entry	Family
**Fungi**					
Ecp11-1	*Passalora fulva*	4 August 2021	X-ray	6ZUS	LARS
APikL2A	*Magnaporthe oryzae*	24 March 2021	X-ray	7NLJ	MAX
APikL2F	*Magnaporthe oryzae*	24 March 2021	X-ray	7NMM	MAX
AVR-PikD	*Pyricularia oryzae*	17 Februrary 2021	X-ray	7BNT	MAX
AVR-PikF	*Pyricularia oryzae*	3 February 2021	X-ray	7B1I	MAX
AVR-PikC	*Pyricularia oryzae*	3 February 2021	X-ray	7A8X	MAX
SnTox3	*Parastagonospora nodorum*	4 November 2020	X-ray	6WES	MAX
Zt-KP6-1	*Zymoseptoria tritici*	4 March 2020	X-ray	6QPK	LysM
MLP124017	*Melampsora larici-populina*	18 December 2019	Solution NMR	6SGO	Cys knot, NTF2-like fold
Mg1LysM	*Zymoseptoria tritici*	16 October 2019	X-ray	6Q40	LysM
AVR-Pia	*Pyricularia oryzae*	10 July 2019	X-ray	6Q76	MAX
AvrPib	*Pyricularia oryzae*	5 September 2018	X-ray	5Z1V	MAX
MlpP4.1	*Melampsora larici-populina*	22 August 2018	Solution NMR	6H0I	Cys knot, NTF2-like fold
Avr4	*Passalora fulva*	22 August 2018	X-ray	6BN0	Chitin-binding
PIIN_05872	*Piriformospora indica*	2 May 2018	X-ray	5LOS	DELD
BEC1054	*Blumeria hordei*	20 June 2018	X-ray	6FMB	RALPH
AVR-PikE	*Pyricularia oryzae*	13 June 2018	X-ray	6G11	MAX
AVR-PikA	*Pyricularia oryzae*	3 June 2018	X-ray	6FUD	MAX
AvrP	*Melampsora lini*	30 August 2017	X-ray	5VJJ	Zn-binding
Avr2	*Fusarium oxysporum*	16 August 2017	X-ray	5OD4	ToxA/TRAF
PevD1	*Verticillium dahliae*	5 July 2017	X-ray	5XMZ	C2-like
Avr4	*Pseudocercospora fuligena*	29 June 2017	X-ray	4Z4A	Chitin-binding
AVR1-CO39	*Magnaporthe oryzae*	14 October 2015	Solution NMR	2MYV	MAX
Prp5	*Saccharomyces cerevisiae*	11 December 2013	X-ray	4LK2	DEAD-box
AvrLm4-7	*Leptosphaeria maculans*	11 December 2013	X-ray	2OPC	LARS
AvrM	*Melampsora lini*	16 October 2013	X-ray	4BJM	RXLR-like
AvrM-A	*Melampsora lini*	16 October 2013	X-ray	4BJN	RXLR-like
Ecp6	*Passalora fulva*	17 July 2013	X-ray	4B8V	LARS
AvrPiz-t	*Pyricularia oryzae*	12 September 2012	Solution NMR	2LW6	MAX
AvrL567-D	*Melampsora lini*	30 October 2007	X-ray	2QVT	RXLR-like
AvrL567-A	*Melampsora lini*	6 March 2007	X-ray	2OPC	RXLR-like
**Oomycetes**					
Avr1d	*Phytophthora sojae*	17 March 2021	X-ray	7C96	RXLR
PsAvh240	*Phytophthora sojae*	6 February 2019	X-ray	6J8L	RXLR/WY
SFI3	*Phytophthora infestans*	5 December 2018	X-ray	6GU1	RXLR/WY
PcRXLR12	*Phytophthora capsici*	15 August 2018	X-ray	5ZC3	RXLR/WY
PSR2	*Phytophthora sojae*	16 August 2017	X-ray	5GNC	RXLR/WY
Avr3a	*Phytophthora infestans*	11 January 2017	Solution NMR	2NAR	RXLR/WY
PexRD54	*Phytophthora infestans*	3 August 2016	X-ray	5L7S	RXLR/WY
ATR13	*Hyaloperonospora parasitica*	18 january 2012	Solution NMR	2LAI	RXLR
AVR3a4	*Phytophthora capsici*	3 August 2011	Solution NMR	2LC2	RXLR
PexRD2	*Phytophthora infestans*	3 August 2011	X-ray	3ZRG	RXLR/WY
Avr3a11	*Phytophthora capsici*	3 August 2011	X-ray	3ZR8	RXLR/WY
ATR1	*Hyaloperonospora parasitica*	20 July 2011	X-ray	3RMR	RXLR/WY

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
