# Peer review of "Exploiting Structural Modelling Tools to Explore Host-Translocated Effector Proteins"

_ijms, 2021, doi:10.3390/ijms222312962_

Round 1

Reviewer 1 Report

The manuscript is intended to be a review of effector proteins involved in plant.microbe interactions. It provides a thorough overview of the role of such proteins, structure modeling tools and at the end uses several examples of modeled structures to assess the applicability of recent methods as AlphaFold2 and RoseTTafold.

In my opinion, the manuscript is overly long in its present form. It basically provides two separate reviews on the proteins and the tools before turning to the message on the specific applicability of the tools for the given case. I suggest a much more concise and focused treatment of the former two topics and I also would like to provide suggestions for improving the presentation of the actual models.

In particular, for structural modeling I would strongly suggest to emphasis the relationship of sequence vs. structure conservation, as structural modeling can significantly contribute to the understanding of these in specific protein families. In addition, the prediction of local structural features like binding sites is of crucial importance in the particular field (see also below). Understanding the mechanism of these proteins is only possible by studying their interactions/complexes in detail, to which structural modeling can greatly contribute.

I also suggest that the authors emphasize the structural classification of the proteins by showing this information in the tables and figures. For example, the structures 4bjm and 2qvt belong to different CATH families etc. In modeling, this can be a major factor, also apparent in the variability of structures shown in Figure 2, that also does not contain any info on the functional and/or structural families. 

For the evaluation of the structural models, I suggest two additional aspects, first, the assessment of the functional (binding) sites in the predicted structures. Also, the authors might want to provide a more specific measure, e.g. TM-score - besides RMSD - on the quality of the structures.

My final suggestion would be the overview of the (possible) effector proteins in the proteomes for which AlphaFold2-predicted proteins are available (https://alphafold.ebi.ac.uk/download). It would be quite interesting to see whether the predicted proteins for which structures are available are accurately modeled, whether there are any effector proteins for which structural modeling (but maybe not sequence similarity alone) can identify evolutionary relatives, or are there effector protein families for which no experimental structure is available yet and how their predicted structure compares to the better characterized families. I am fully aware that this can be a lot of work and might not be feasible within the time of the revision, but it would perfectly align with the aim of the review (and what its title promises).

Minor remarks:

I would recommend to make the title of the paper more specific instead of mentioning 'effector proteins' (phrase too general).

I recommend that the manuscript is substantially rewritten for the sake of clarity. For example, Line 358 repeats the info (with more precise phrasing) already present in line 290. I do not think that in such a review a textbook-style introduction to structural modeling/structure determination is in place. It is very hard to write an accurate description of these processes, it gives room to many imprecise phrasings, and in the present manuscript there are also some points which are not phrased well, for example:

Line394 should be rephrased, AlphaFold2 is completely redesigned pipeline compared to the first version, not a simple improvement.
Line 309: "allows for a good dynamic information on the protein" - this seems like a phrasing by a non-expert, should be rewritten (or I would prefer to omit all the details of the structure determination methods in order to avoid similar "naive" statements).

Please make sure to use all terms uniformly:

Peronosporales not italic (line 122), Pernosporales - typo (l. 171)
alphafold is written in lowercase several times - should be uniformly AlphaFold2

Typos/minor things (only a highlight):
Line 140: "The effector domain of [organism]" should include the protein containing the domain 
Line 190: more intensiveLY
Line 326: "due to the newness"
Line 473: lDDT is "Local Distance Difference Test" (node "Different test"),
see https://www.ncbi.nlm.nih.gov/pmc/articles/PMC3799472/
Line 508: inadvertent line break

The abstract is provided in place of "Correspondence" on the first page

Author Response

Reviewer 1:

The manuscript is intended to be a review of effector proteins involved in plant.microbe interactions. It provides a thorough overview of the role of such proteins, structure modeling tools and at the end uses several examples of modeled structures to assess the applicability of recent methods as AlphaFold2 and RoseTTafold.

  • In my opinion, the manuscript is overly long in its present form. It basically provides two separate reviews on the proteins and the tools before turning to the message on the specific applicability of the tools for the given case. I suggest a much more concise and focused treatment of the former two topics and I also would like to provide suggestions for improving the presentation of the actual models.

Thank you for the comment. We have condensed some parts of the manuscript’s texts on different effector proteins focusing more on translocated effector proteins rather than apoplastic effector proteins. As suggested by reviewer 1, major changes have been implemented in chapter 1 and 2. Because of the major changes, we have decided to exclude chapter 2 from the track changes.

These major changes also included deletion and moving some parts of the text that allowed us to make the text more concise and include text incorporated in response to reviewer comments.

  • In particular, for structural modeling I would strongly suggest to emphasis the relationship of sequence vs. structure conservation, as structural modeling can significantly contribute to the understanding of these in specific protein families. In addition, the prediction of local structural features like binding sites is of crucial importance in the particular field (see also below). Understanding the mechanism of these proteins is only possible by studying their interactions/complexes in detail, to which structural modeling can greatly contribute.

This is indeed a very important point. However, there are not many studies done in relation with effector protein target sites and their sequence-structure relationship. We have implemented some text indicating this matter and emphasize on the fact that computational modelling could be great alternatives to time and energy consuming experimental methods in investigating potential binding sites in microbe-plant interactions. These changes can be seen in lines 479-484.

  • I also suggest that the authors emphasize the structural classification of the proteins by showing this information in the tables and figures. For example, the structures 4bjm and 2qvt belong to different CATH families etc. In modeling, this can be a major factor, also apparent in the variability of structures shown in Figure 2, that also does not contain any info on the functional and/or structural families. 

We appreciated this suggestion; Table 2 has been updated with information about the protein families of different effector proteins. Some effector proteins are not grouped in a particular effector family; hence, we decided to add some special features of these proteins in the table.

There is also some information added about the functional and structural patterns of each effector protein existing in Figure2 in line 406-439.

  • For the evaluation of the structural models, I suggest two additional aspects, first, the assessment of the functional (binding) sites in the predicted structures. Also, the authors might want to provide a more specific measure, e.g. TM-score - besides RMSD - on the quality of the structures.

Thank you for your comment. The emphasized aim of this review was to compare and introduce the newly released computational modelling procedures namely, Rosettafold and AlphaFold2 Monomer, which has only been released before this review.  Details on experimental biochemical, structural and modelling approaches of effector-target interaction is rather limited. However, investigating the binding sites of effector proteins is a novel and comprehensive field of study, which is more suitable for a research paper. Some modelling approaches namely AlphaFold-Multimer are capable of studying the potential interactions of the effector proteins and their targets. AlphaFold-Multimer has been named in the review in Line 481.

We have implemented GDT-TS score as a more specific measure, besides RMSD, to assess the quality of the modelled structures. For better understanding the RMSD values are shown in blue and GDT-TS values in red.

  • My final suggestion would be the overview of the (possible) effector proteins in the proteomes for which AlphaFold2-predicted proteins are available (https://alphafold.ebi.ac.uk/download). It would be quite interesting to see whether the predicted proteins for which structures are available are accurately modeled, whether there are any effector proteins for which structural modeling (but maybe not sequence similarity alone) can identify evolutionary relatives, or are there effector protein families for which no experimental structure is available yet and how their predicted structure compares to the better characterized families. I am fully aware that this can be a lot of work and might not be feasible within the time of the revision, but it would perfectly align with the aim of the review (and what its title promises).

Thank you for your comment. We have overviewed the possible effector proteins that are already available in the AlphaFold2 prediction database in the https://alphafold.ebi.ac.uk/download website. The only models available for effector proteins was from Saccharomyces cerevisiae species, which according to the information existing until now, PrP5 is the only effector protein structured and revealed in PDB. As this effector protein was the only effector protein available in this database, we have avoided using this data.

Minor remarks:

  • I would recommend to make the title of the paper more specific instead of mentioning 'effector proteins' (phrase too general).

As recommended by the reviewer, we have modified the title.

  • I recommend that the manuscript is substantially rewritten for the sake of clarity. For example, Line 358 repeats the info (with more precise phrasing) already present in line 290. I do not think that in such a review a textbook-style introduction to structural modeling/structure determination is in place. It is very hard to write an accurate description of these processes, it gives room to many imprecise phrasings, and in the present manuscript there are also some points which are not phrased well, for example:

Line394 should be rephrased, AlphaFold2 is completely redesigned pipeline compared to the first version, not a simple improvement.
Line 309: "allows for a good dynamic information on the protein" - this seems like a phrasing by a non-expert, should be rewritten (or I would prefer to omit all the details of the structure determination methods in order to avoid similar "naive" statements).

Thank you for the comment. We have checked the text for probable repetitions and implemented some changes and, in some cases, deleted some parts. Lines 280-284, the writing about different modelling procedures was omitted and retained this in lines 378-382.

Line 394 is rephrased emphasizing the fact that AlphaFold2 was a breakthrough in comparison to AlphaFold1by using distinguished transformers and repeated loops of analysis in line 314.

The modelling procedures and their mode of action is a highly specialized area of bioinformatics. We intended to explain the complex procedures in a simple way with simple phrases in order to welcome a broad range of readers especially non-specialized researchers in the modelling field.

  • Please make sure to use all terms uniformly:

Peronosporales not italic (line 122), Pernosporales - typo (l. 171)
alphafold is written in lowercase several times - should be uniformly AlphaFold2

Thank you for the comment, terms have been unified.

  • Typos/minor things (only a highlight):
    Line 140: "The effector domain of [organism]" should include the protein containing the domain 
    Line 190: more intensiveLY
    Line 326: "due to the newness"
    Line 473: lDDT is "Local Distance Difference Test" (node "Different test"),
    see https://www.ncbi.nlm.nih.gov/pmc/articles/PMC3799472/
    Line 508: inadvertent line break

Thank you for the comments. All above changes have been implemented.

  • The abstract is provided in place of "Correspondence" on the first page

Thank you for the attention to details; the abstract has been placed in the correct position.

Reviewer 2 Report

The review is well written and covers the topic extensively. Morover it contain comparison of structure prediction tools performance against published PDB structures.

I have some comments:

Figure 1 was not included in the pdf file for review

Figure 2 Was is checked whether the pdb structures were not included in the training set for APlphafold? SPecificily those released earlier (like Avr2 released in 2017)

line 345 you mean that the absence of structures of some proteins in Table 2 highlights the claimed statement?

Author Response

Reviewer 2:

The review is well written and covers the topic extensively. Morover it contain comparison of structure prediction tools performance against published PDB structures.

I have some comments:

  • Figure 1 was not included in the pdf file for review

We are sorry for the inconvenience. We have sent the PDF document of Figure 1, but it was not displayed in the manuscript. We are not aware of the reason. We will send the PDF again and check the submission for the inclusion of the figure.

  • Figure 2 Was is checked whether the pdb structures were not included in the training set for APlphafold? SPecificily those released earlier (like Avr2 released in 2017)

Thank you for the suggestion. Yes, it was checked and the date of using the PDB templates were tested with both before and after the published template date of each effector protein, no significant difference have been seen.

  • line 345 you mean that the absence of structures of some proteins in Table 2 highlights the claimed statement?

Thank you for the comment. Table2 provides the known structures of filamentous effector proteins at present. As mentioned in the upper part of the text, RxLR effector proteins are more represented in contrast to other types of effector proteins, which with the new edits of Table2 indicating the effector family types is now more visible.

Round 2

Reviewer 1 Report

I thank the authors for all they efforts in improving the manuscript according to my comments. I have just one very small request (and apologies I did not indicate this before): kindly rephrase the sentence on lines 270-271 and also the heading to Table 2 to avoid using the term "PDB" in the sense "experimentally determined structure" / "PDB-deposited structure" (or similar). I hope this is a minor modification that can be done in the proof stage.